# Risk Factors for Eating Disorders in University Students: The RUNEAT Study

**DOI:** 10.3390/healthcare12090942

**Published:** 2024-05-04

**Authors:** Imanol Eguren-García, Sandra Sumalla-Cano, Sandra Conde-González, Anna Vila-Martí, Mercedes Briones-Urbano, Raquel Martínez-Díaz, Iñaki Elío

**Affiliations:** 1Research Group on Foods, Nutritional Biochemistry and Health, Universidad Europea del Atlántico, 39011 Santander, Spain; imanol.eguren@uneatlantico.es (I.E.-G.); sandra.sumalla@uneatlantico.es (S.S.-C.); sandra.conde@alumnos.uneatlantico.es (S.C.-G.); mercedes.briones@unib.org (M.B.-U.); raquel.martinez@uneatlantico.es (R.M.-D.); 2Department of Health, Nutrition and Sport, Universidad Internacional Iberoamericana, Campeche 24560, Mexico; 3Research Group M3O, Methodology, Methods, Models and Outcomes of Health and Social Sciences, Faculty of Health Sciences and Welfare, University of Vic-Central University of Catalonia, 08500 Vic, Spain; anna.vilamarti@uvic.cat; 4Institute for Research and Innovation in Life Sciences and Health in Central Catalonia (IRIS-CC), 08500 Vic, Spain; 5Department of Health, Universidad Internacional Iberoamericana, Arecibo, PR 00613, USA; 6Faculty of Health Sciences, Universidade do Cuanza, Cuito EN250, Bié, Angola; 7Faculty of Health Sciences, Universidad de La Romana, La Romana 22000, Dominican Republic

**Keywords:** eating disorders, first and fourth year of university, adherence to Mediterranean diet, emotional intake, body composition

## Abstract

The purpose of the study is to assess the risk of developing general eating disorders (ED), anorexia nervosa (AN), and bulimia nervosa (BN), as well as to examine the effects of gender, academic year, place of residence, faculty, and diet quality on that risk. Over two academic years, 129 first- and fourth-year Uneatlántico students were included in an observational descriptive study. The self-administered tests SCOFF, EAT-26, and BITE were used to determine the participants’ risk of developing ED. The degree of adherence to the Mediterranean diet (MD) was used to evaluate the quality of the diet. Data were collected at the beginning (T1) and at the end (T2) of the academic year. The main results were that at T1, 34.9% of participants were at risk of developing general ED, AN 3.9%, and BN 16.3%. At T2, these percentages were 37.2%, 14.7%, and 8.5%, respectively. At T2, the frequency of general ED in the female group was 2.5 times higher (OR: 2.55, 95% CI: 1.22–5.32, *p* = 0.012). The low-moderate adherence to the MD students’ group was 0.92 times less frequent than general ED at T2 (OR: 0.921, 95%CI: 0.385–2.20, *p* < 0.001). The most significant risk factor for developing ED is being a female in the first year of university. Moreover, it appears that the likelihood of developing ED generally increases during the academic year.

## 1. Introduction

Eating disorders (ED) are a group of serious mental illnesses, characterized by alterations in food intake due to a distorted perception of body image and an excessive preoccupation with body image and/or food eaten [1]. ED covers a wide spectrum of clinical manifestations, the most well-known of which are anorexia nervosa (AN) and bulimia nervosa (BN). Anorexia nervosa is defined as the voluntary limitation of food intake induced by a strong fear of weight gain and a distortion of body self-perception, resulting in an exceedingly low body weight. BN is distinguished by alternating bouts of loss of control over eating, recurrent episodes of binge eating, and the use of compensatory behaviors such as self-induced vomiting, enemas, diuretics, and other drugs [2]. Although anyone can develop an eating disorder, the largest incidence rates occur between the ages of 15 and 19, particularly among women [3].

University students are a high-risk population for developing ED [4]. Validated screening questionnaires are used to determine the risk of ED in the population under study. A result above the predefined cutoff value for each test indicates that the individual is at risk of developing ED.

A recent meta-analysis of data from 145,629 university students found that 19.7% are at risk of developing ED [5]. Academic stress may explain the rise in the number of cases of ED and other mental health issues during university study [6]. Eisenberg et al. [7] observed an increase in the risk of developing ED from 9% to 13% for females and from 3% to 4% for men in a sample of 2822 university students two years after starting their studies.

Furthermore, a recent study found a significant increase in the number of ED among both male and female university students from 2009 to 2021 [8]. Several studies in Spain have looked into the likelihood of acquiring an eating disorder among university students. In a study of 1306 university students who completed the SCOFF questionnaire, Martínez-González et al. [4] identified 21.2% of females and 15% of males as at risk of having ED. In a study of 422 university students, Castelao–Naval et al. [9] showed that 12.8% were at risk of developing AN (evaluated by the EAT-26 questionnaire) and 4.7% of BN (measured by the BITE questionnaire).

In 2020, the University of Castilla–La Mancha conducted a cross-sectional study with a sample of 481 university students and found that the risk of developing ED was 17.4% in the case of males and 32.4% in the case of females using the SCOFF questionnaire [10]. Finally, Alcaraz–Ibañez et al. [11] found a prevalence of 22.7% in a sample of 545 Spanish university students measured by the SCOFF questionnaire.

The start of university studies is a moment of significant lifestyle change, particularly in eating. Students frequently reduce the quality of their food, leading to alterations in body image [12,13]. This effect is especially prominent among students living in student housing [14], who engage in more frequent binge eating behaviors and are at a higher risk of developing ED than those who stay at home [4].

The Mediterranean diet (MD) is a well-balanced dietary pattern characterized by a high consumption of vegetables, fruit, cereals, legumes, nuts, and olive oil, a moderate consumption of fish, eggs, and dairy products, and a low intake of meat, animal fats, and processed foods. It is regarded as a benchmark of healthy eating; thus, adhering to it helps to preserve excellent health and prevent the advent of non-communicable chronic diseases, including ED [15]. However, Spanish university students do not often adhere to the MD, which is more prevalent in the south [16] than in the north [17]. Although we believe that maintaining high adherence to the MD may reduce the chance of developing ED in the university population, this association has not previously been studied, and evidence is limited.

There is little research that assess the relative risk of developing ED among different university faculties. Banna et al. [18] found no difference between university students from different faculties in terms of the risk of developing ED. It is unclear whether the danger is larger among incoming or final-year undergraduates. Tavolacci et al. [19] discovered, however, that girls in first- and second-year social sciences majors were more likely to have ED.

According to Iyer and Shriraam [20] and Attouche et al. [21], health sciences students are overly concerned with controlling their health status and diet, increasing the risk of developing ED. However, it is also a group of students who are more likely to follow nutritional guidelines, whereas poorer eating habits are related to higher results on ED screening tests [4,22].

Most previous research has only examined the overall prevalence of the risk of developing ED, without determining which categories of university students are most likely to have ED [5,7,9,10,11]. Identifying these students is a critical step toward establishing effective prevention interventions. It is also urgent, considering the worrisome rise in the number of cases.

Although the research problem has not been studied in an integrated manner in the university population, based on the existing bibliography for each of the main variables under study (gender, place of residence, academic year, faculty, and diet quality), the following hypotheses are established.

Compared to men, women are more likely to develop ED. Independent students are more prone to develop an eating disorder since they have control over their nutrition. When faced with considerable lifestyle and behavioral changes, first-year students are more likely than fourth-year students to develop an eating disorder. Students pursuing health sciences are more likely to develop eating disorders since they are more concerned about their own health. Higher adherence to the MD is connected with a lower chance of developing ED among students.

Despite the topic’s limited bibliographical basis, the current study’s goal is to analyze the risk of developing general ED, AN, and BN, as well as to investigate the impact of gender, academic year, place of residence, faculty, and diet quality on that risk.

## 2. Materials and Methods

### 2.1. Study Design

This was an observational descriptive study that assessed the likelihood of developing general ED, AN, and BN in university students at the beginning and at the end of an academic year and its relationship with socio-demographic and lifestyle factors.

The study was approved by the Ethics Committee of the Universidad Europea del Atlántico, which followed the ethical standards established in the Declaration of Helsinki, with code (CEI-11/2018). The protocol for monitoring first-year university students eating behavior is also available [23].

### 2.2. Participant Recruitment

Research was conducted at the Universidad Europea del Atlántico during two different academic years (2018/2019 and 2021/2022), due to the interruption caused by the COVID-19 pandemic. The inclusion criteria were to be studying for a degree at the Universidad Europea del Atlántico, to be a first- or fourth-year student, and to be available to perform the measurements at the beginning and end of the academic year. Exclusion criteria were being an undergraduate student in a course other than the first or fourth year, being a master’s or doctoral student, being a temporary resident student, or having a diagnosed ED.

The sample size was calculated for finite populations by counting the students who enter the first year at the Universidad Europea del Atlántico, it was calculated with a 90% confidence level, a 5% error range, a precision level (d = 3%), and an expected proportion of losses of 15%. The total size of the Universidad Europea del Atlántico (*n* = 400 students) and the representative sample size (*n* = 124 students).

Volunteers were recruited using a non-probability sampling strategy. At the beginning of each academic year, informative posters were hung up and first- and fourth-year classes were visited to explain the participation in the study. Contact details of the volunteers were collected (name, surname, and institutional email address) after they signed an informed consent form for participation in the study.

The sample, after eliminating those subjects who did not complete each of the phases of the study, consisted of a total of 129 students, 70 from the 2018/2019 academic year and 59 from the 2021/2022 academic year (Figure 1).

### 2.3. Data collection and measures

The study had two identical stages, one at the beginning of the academic year in September (T1) and one at the end in June (T2):First stage: Administration of an online survey: We sent each participant a link to a Google Forms questionnaire and a numerical code to maintain anonymity. The survey asked about socio-demographic variables (gender, date of birth, place of birth, grade, place of residence) and administered the following questionnaires:
-Eating attitude test (EAT-26): This is the abbreviated version of a self-administered questionnaire used globally to assess behavior and attitudes towards food present in AN. It was validated in the Spanish population in 2010 [24]. It consists of 26 items, where a total score above 20 points is related to a risk of developing AN with a sensitivity of 59.74% and a specificity of 94.94%.-Bulimia investigatory test Edinburgh (BITE): This assesses the presence and severity of symptoms related to BN. It consists of 33 items with a sensitivity of 93.0% and a specificity of 55% and is interpreted as follows: If the final score is <10 points (no bulimia symptoms); between 10–20 points (presence of abnormal eating patterns); >20 points (presence of abnormal eating patterns with the possibility of developing BN) [25].-Eating behavior test (SCOFF): This is a short and simple questionnaire consisting of 5 dichotomous questions that detect possible ED. Each affirmative answer scores 1 point, and the questionnaire is considered positive when the person answers affirmatively to two or more questions. The questionnaire provides a sensitivity of 100% for AN and BN and a specificity of 87.5% [26].-Emotional eating questionnaire (EEQ): This is a questionnaire designed to assess the influence of emotions on eating. It consists of a total of 10 questions and is interpreted as follows: 0–5 points (non-emotional eater); 5–10 points (less emotional eater); 10–15 points (emotional eater); 15–20 points (very emotional eater) [27].-The MEDAS-14 (MEDAS-14): This is a validated dietary tool used in the PREDIMED study to assess the degree of adherence to a pattern of the MD, which is considered a high-quality diet standard. It consists of a total of 14 items and is interpreted as follows: >8 points (good adherence); 5–8 points (improving adherence); and <5 points (low adherence) [28].
Second stage: Once they completed the survey, we contacted participants to make an appointment at the Nutrition Care Centre to perform anthropometric measurements. The INBODY 720^®^ electrical bioimpedance machine was used to determine body weight, amount and percentage of body fat, fat-free mass, and total body water. Days before attending the nutritional consultation, participants were explained the standardization considerations before performing an electrical bioimpedance test so that the measurement would be correct. Height was determined using a GIMA^®^ measuring rod with an accuracy of 0.5 cm. The weight and height data were then used to calculate the body mass index (BMI), using the World Health Organization (WHO) BMI classification [29].

### 2.4. Statistical Analysis

The normality distribution of quantitative variables was assessed using the Shapiro–Wilks normality test. We expressed quantitative variables as means and standard deviations (mean ± SD). We expressed qualitative variables as absolute (*n*) and relative (%) frequencies. For the contrast of differences between groups, we used the parametric Student’s *t*-test for independent samples, and to compare the means of two measurements taken from the same individuals at two different times, we used Student’s *t*-test for paired samples.

The determination of relationships between qualitative variables was done using Pearson’s Chi-square test and Fisher’s exact test when a cell had an expected frequency lower than 5. McNemar tests were used for paired nominal data when comparing the same group of students at the beginning and end of the academic year. The odds ratio (OR) was used as a measure of association, with a 95% confidence interval.

For all hypothesis tests, we used a 95% confidence interval and a bilateral statistical significance *p*-value of <0.05. All computer data were analyzed using Jamovi software (version 2.2.5) for Windows.

## 3. Results

### 3.1. Sociodemographic Variables and Anthropometric Variables

The socio-demographic data are shown in Table 1. The sample consisted of a total of 129 students, of whom 67 (51.9%) were males and 62 (48.1%) were females. The mean age was 20.3 ± 2.2 years. Of the students, 34.9% were independent and 65.1% still resided at home with their parents. Additionally, 80.6% were in their first year of university, 78.3% were studying for a degree in the faculty of health sciences, and 95.3% were from Spain.

Table 2 shows the measurements of anthropometric variables at the beginning of the academic year in September (T1) and at the end of June (T2). Men significantly increased their body weight throughout the course (74.6 kg vs. 76.0 kg; *p* < 0.001), though this was not so in the women’s group. Regarding BMI, at T1, 19.4% of the students were in the overweight/obese category, with significantly more males than females (26.9% vs. 11.3%, *p* = 0.025). At T2, the general percentage of overweight/obese students increased slightly (20.2%), with significantly more males than females (31.3% vs. 8.1%, *p* = 0.025). On the other hand, both men and women significantly increased their amount of fat-free mass (64.1 kg vs. 65.0 kg, *p* = 0.005; 42.5 kg vs. 43.1 kg, *p* = 0.001) and total body water (46.9 kg vs. 48.3 kg, *p* < 0.001; 31.1 kg vs. 31.5 kg, *p* = 0.001). Finally, there were no significant changes in the amount or percentage of body fat in either the male or female groups.

Both men and women significantly increased their amount of fat-free mass (64.1 kg vs. 65.0 kg, *p* = 0.005; 42.5 kg vs. 43.1 kg, *p* = 0.001) and total body water (46.9 kg vs. 48.3 kg, *p* < 0.001; 31.1 kg vs. 31.5 kg, *p* = 0.001). Finally, there were no significant changes in the amount or percentage of body fat in either the male or female groups.

### 3.2. Risk of Developing Eating Disorders, Degree of Adherence to Mediterranean Diet, and Influence of Emotions at Mealtimes Compared at the Beginning and the End of the Course

Table 3 shows the results of the different ED risk, MEDAS-14, and EEQ tests. There was a significant decrease in students at risk of developing BN (BITE test) between T1 and T2 (16.3% vs. 14.7%, *p* < *0*.001).

Regarding the risk of developing AN (EAT-26 test), there was an increase in risk between T1 and T2, although this increase was not significant (3.9% vs. 8.5%). In contrast, there was no significant difference in the overall risk of developing general ED (SCOFF test) between T1 and T2 (34.9% vs. 37.2%).

In terms of adherence to the MD, there was a significant increase in the number of students who achieved good adherence to the MD at T2 (21.7% vs. 35.7%, *p* = 0.008). There was also no difference in the influence of EEQ at T1 and T2 (46.5% vs. 41.1%).

### 3.3. Risk of Developing General Eating Disorders at the Beginning and the End of the Course and Associated Factors

Fourth-year students had a lower prevalence (12% vs. 48.1%) and were 0.15 times more likely to develop ED at T1 (OR: 0.147, 95%CI: 0.0415–0.522, *p* = 0.001), and at T2 they had had a lower prevalence (20% vs. 41.3%) and were 0.05 times less likely to develop ED (OR: 0.355, 95%CI: 0.124–1.02, *p* = 0.047).

Furthermore, at T1, groups of students from the faculties of social sciences and humanities (SSH) and higher polytechnic school (HPS) had a higher prevalence (57.1% vs. 36.6%) and were 2.3 times more likely to develop of ED (OR: 2.31, 95%CI: 0.985–5.40, *p* = 0.051) and students whose emotions were very influential had a higher prevalence (50% vs. 33.3%) and were 2 times more likely to develop ED (OR: 2.00, 95%CI: 0.982–4.08, *p* = 0.055), although these differences were close to being statistically significant.

At T2, the female group had a higher prevalence (26.9% vs. 48.4%) and was 2.5 times more likely to develop ED (OR: 2.55, 95%CI: 1.22–5.32, *p* = 0.012). In addition, the group with a low-moderate adherence to the MD had a lower prevalence (24.1% vs. 60.9%) and was 0.92 times less likely to develop ED (OR: 0.921, 95%CI: 0.385–2.20, *p* = <0.001) (Table 4).

### 3.4. Risk of Developing Anorexia Nervosa at the Beginning and the End of the Course and Associated Factors

When analyzing the association of developing AN (EAT-26 test) in Table 5, the female group had a higher prevalence in T1 (8.1% vs. 0%) and was 12.9 times more likely to develop AN (OR: 12.9, 95%CI: 0.69–239, *p* = 0.024), and in T2 (14.5% vs. 3%), the group was 5.5 times more likely to develop AN (OR: 5.52, 95%CI: 1.14–26.6, *p* = 0.026).

In addition, at T2, a high adherence to the MD (MEDAS-14 test) had a higher prevalence (15.2% vs. 4.8%) and was 3.5 times more frequent than AN (OR: 3.54, 95%CI: 0.979–12.8, *p* = 0.043). No significant differences were observed with respect to the variables of place of residence, academic year, or faculty.

### 3.5. Risk of Developing Bulimia Nervosa at the Beginning and the End of the Course and Associated Factors

Table 6 shows the risk of BN (BITE test) both at T1 and T2. The female group had a higher prevalence in T1 (23% vs. 9%) and was 3 times more likely to develop BN (OR: 3.03, 95%CI: 1.08–8.48, *p* = 0.049), and SSH + EPS had a higher prevalence (28.6% vs. 12%) and was 2.9 times more likely to develop BN (OR: 2.93, 95%CI: 1.06–8.12, *p* = 0.033). In addition, students whose emotions were very influential had a higher prevalence (26.7% vs. 5.9%) and were 5.8 times more likely to develop BN (OR: 5.82, 95%CI: 1.82–18.6, *p* = 0.001).

At T2, although the difference was not statistically significant, the trend remained that the female group had a higher prevalence (21% vs. 9%) and was 2.7 times more likely to develop BN (OR: 2.70, 95%CI: 0.955–7.61, *p* = 0.054). On the other hand, students whose emotions were very influential had a higher prevalence (22.6% vs. 9.2%) and were 2.9 times more likely to develop BN (OR: 2.89, 95%CI: 1.05–7.91, *p* = 0.034).

## 4. Discussion

The percentage of students at risk of developing general ED in our study at the beginning of the academic year was 34.9% and increased to 37.2% at the end of the course, as determined through the administration of the SCOFF test. These results are superior to the data found in the meta-analysis of Alhaj et al. [5], which determined that the overall prevalence risk of general ED in university students is 27.6%, using the SCOFF test. However, if we limit the comparison to data from Spain, our results are more consistent with the meta-analysis (31.7%).

Using the EAT-26 test, we were able to determine that, more precisely, the probability of developing AN in our group increased from 3.9% at the start of the study to 8.5% at the end of the academic year. In this specific case, the risk percentages are less than the median prevalence determined by Alhaj et al. [5] (17.0%) and other Spanish universities [9] (12.8%), with the EAT-26 test being used to measure these percentages.

Finally, the risk of developing BN in our sample was 16.3% at the beginning of the year and was the only risk variable whose prevalence decreased at the end of the school year [14.7%], using the BITE test. In contrast to the previous cases, we found a higher prevalence of risk of developing BN than in other Spanish universities, such as Castelao–Naval et al. [9], who found a prevalence of risk of 4.7%.

In the study of the risk variables analyzed, in general we found a higher prevalence of ED within the female group, which is consistent with results at other Spanish universities [4,5,6,7,8,9,11], regardless of the test used. Other universities in Europe [30] and Malaysia [31] also have a higher risk of developing ED among female university students. Our study reinforces the hypothesis that females have a higher risk of developing not only general ED, but also AN and BN. Leaving aside the biological differences between males and females, a study conducted at the University of Cordoba [32] revealed that university females have lower levels of body satisfaction, lower self-esteem, poorer self-perception, and a desire for a slimmer body. All these socio-emotional factors correlate with a disordered eating attitude, which may explain why females at universities have a higher risk of developing ED.

Regarding residence, we found no distinction between students who live at home and those who have become independent in the prevalence of the risk of developing general ED, AN, or BN. This finding opposes that found by Martínez–González et al. [4], who found that students who lived in shared flats had a higher probability of developing ED. This result contradicts our hypothesis since we believed that independent students, being responsible for their nutrition, could present a greater alteration of eating habits, especially those who were in their first year away from home, with the consequent appearance of ED.

New students are more likely to experience general ED during the academic year than more specialized conditions like AN or BN. This is one of the variables that shows the greatest statistically significant variations at the start and finish of the research. According to other research [19], first- and second-year students are twice as likely to experience problems than students in subsequent years. The first year of university is a time of significant lifestyle and socioemotional changes, with the potential rise in ED and feelings of dissatisfaction with physical appearance. The results are consistent with our initial premise. Further research would be required to support this theory, though, as many studies [33,34,35] only gather or present data from the first year and neglect to track the change in the risk of developing ED during university education.

Thirdly, we discovered that although this difference was not statistically significant, individuals pursuing degrees unrelated to the health sciences were more likely to start the academic course with general ED. Furthermore, we found that these same students had an increased chance of developing BN, although, by the end of the research, this difference had stopped being significant. While Harris et al. [36] found no significant differences between nutrition and sports science majors and other majors outside the health sciences, Martínez–González et al. [4] and Tavolacci et al. [19] report similar results.

In larger terms, this student profile indicates a higher interest in mental health, physical activity, and diet, all of which lower the chance of developing ED. Nevertheless, these studies may occasionally result in an unhealthy obsession with food, exercise, and body image, which raises the risk of developing eating disorders (ED). However, the majority of the students in that study from the faculty of social sciences and humanities were majoring in journalism and audiovisual communication. Because they are more exposed to the media than other students, they are more likely to compare their physical appearance to professionals in traditional and social media, which increases body dissatisfaction—one of the main trigger factors for EDs—and may explain why they have a higher risk of developing EDs [37].

In terms of MD adherence, at the beginning of the course, only 21.7% of the students had high adherence, but at the end of the academic year, this number had considerably increased to 35.7%. Our students’ adherence to the MD is higher than that of other research done in Spanish universities in the north and center of the country, where the range of students with high adherence to the MD is 3.6%–28.4% [17,38,39,40]. The exception to this is a study by Zurita–Ortega et al. [16], where 77.6% of students had high adherence to the MD. However, it should be noted that this research was done in southern universities, where there is a greater representation of Mediterranean culture.

In relation to ED, we found that at the end of the course, those students with higher adherence to the MD had a higher risk of developing general ED and AN. These results reject our initial hypothesis that maintaining a high-quality diet is a protective factor against the development of chronic diseases, including mental illnesses [15]. This may be due to people who are more concerned about their weight and appearance trying to follow more strictly healthy eating patterns such as MD. Further research will be needed to prove this new hypothesis.

Ultimately, we aimed to evaluate the connection between the level of emotions during mealtimes and the likelihood of developing ED. At the start and finish of the course, we discovered a statistically significant positive correlation for BN. Emotional disturbances during mealtimes may be the cause of binge eating episodes, although there is not enough evidence to support this theory.

Our study is innovative in that we used multiple tests to assess the risk of developing general ED, AN, and BN at the start and end of an academic course. We then compared the influence of various categorical variables for each of them in order to identify common variables that contribute to the establishment of a profile of university students who are more likely to develop ED. The basis for creating more specialized health intervention programs to lower the prevalence of these diseases in the university population is identifying common characteristics among university students who are at risk of acquiring ED. It is also among the first studies to evaluate the connection between the risk of developing ED in Spain’s university population and MD adherence.

However, our study also has some limitations.

The first limitation is that this was an initial study to determine whether the categorical variables gender, place of residence, academic year, and faculty have an effect on the risk of developing eating disorders in university students; therefore, no firm conclusions can be extrapolated from this study. In addition, as this was the first exploratory study and the effect of the individual variables was unknown, no statistical study was performed to evaluate the effect of the variables as a whole.

The second limitation is that the first- and fourth-year groups are made up of different students, so it was not possible to assess the effect of how ED risk evolves for the same cohort throughout their four-year degree. Furthermore, there was a high lack of commitment from the participants to complete all phases of the study and a high dropout rate, in addition to the fact that the number of first-year students is significantly higher than the number of fourth-year students.

The third limitation is that only a sample of Spanish university students was used, so the results may not be generalizable to the international university community. It would be desirable to replicate the study at other universities outside of Spain.

The last limitation is that we assessed the risk of developing ED using specific tests, rather than a more objective method such as an interview with a psychiatrist, so a positive screening test result does not confirm that the student has an ED. In addition, although not valued in our study, the repeated measures design may contribute to problems of social desirability, where participants may modify their responses to obtain the expected results and please the researchers.

## 5. Conclusions

Based on the foregoing, we may conclude that being a female in the first year of university is the most important factor in increasing the likelihood of developing ED. Furthermore, it appears that the overall risk of having an eating disorder increases during the school year. However, further research is needed to assess the impact of academic affiliation and site of residence. Furthermore, it is unclear whether following a high-quality diet, such as the MD, increases the risk of developing ED.

Some common traits among students at higher risk of developing ED can be identified, although these findings must be validated in other investigations. These traits can therefore serve as a foundation for future research investigations and the development of more focused food education campaigns to prevent these problems in this population.

## Figures and Tables

**Figure 1 healthcare-12-00942-f001:**
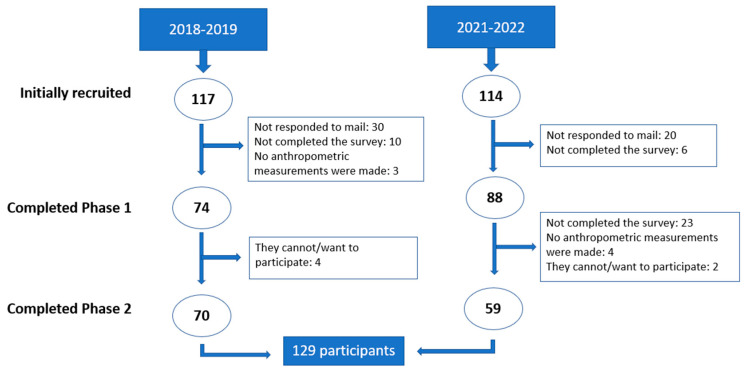
Flow chart of the participant selection process.

**Table 1 healthcare-12-00942-t001:** Sociodemographic characteristics of the participants.

	Males (*n* = 67)	Females (*n* = 62)	Total (*n* = 129)	*p*-Value
Age (Year)	20.9 ± 2.4	19.7 ± 1.7	20.3 ±2.2	0.001 ^1^
Place of residence N (%)				
Independent	23 (34.3)	22 (35.5)	45 (34.9)	0.891 ^2^
Family home	44 (65.7)	40 (64.5)	84 (65.1)	
Course N (%)				
First	52 (77.6)	52 (83.9)	104 (80.6)	0.369 ^2^
Fourth	15 (22.4)	10 (16.1)	25 (19.4)	
Faculty N (%)				
HS	55 (82.1)	46 (74.2)	101 (78.3)	0.277 ^2^
SSH + HPS	12 (17.9)	16 (25.8)	28 (21.7)	
Birthplace N (%)				
Spain	65 (97.0)	58 (93.5)	123 (95.3)	0.427 ^3^
Latin America	2 (3.0)	4 (6.5)	6 (4.7)	

Health and sciences (HS), social sciences and humanities (SSH), higher polytechnic school (HPS). ^1^ T student independent sample (*p* < 0.05); ^2^ Pearson Chi-square (*p* < 0.05); ^3^ Fisher’s exact test (*p* < 0.05).

**Table 2 healthcare-12-00942-t002:** Anthropometric variables of participants at the beginning and end of the course.

	Males (*n* = 67)	Females (*n* = 62)	*p*-Value	Total (*n* = 129)
Weight (kg)				
T1	74.6 ± 9.85	58.3 ± 8.63	<0.001 ^1^	66.8 ± 12.3
T2	76.0 ± 9.51	58.4 ± 9.03	<0.001 ^1^	67.5 ± 12.8
*p*-value	<0.001 ^2^	0.969 ^2^		0.172 ^2^
Height (m)				
T1	1.79 ± 0.07	1.63 ± 0.07	<0.001 ^1^	1.72 ± 0.1
T2	1.79 ± 0.07	1.63 ± 0.07	<0.001 ^1^	1.72 ± 0.1
*p*-value	--	--		--
BMI (kg/m^2^)				
T1				
Normal weight (18.5–24.9)	49 (73.1%)	55 (88.7%)	0.025 ^3^	104 (80.6%)
Overweight/obesity (>25.0)	18 (26.9%)	7 (11.3%)		25 (19.4%)
T2				
Normal weight (18.5–24.9)	46 (68.7%)	57 (91.9%)	0.025 ^3^	103 (79.8%)
Overweight/obesity (>25.0)	21 (31.3%)	5 (8.1%)		26 (20.2%)
*p*-value	0.317 ^4^	0.414 ^4^		0.796 ^4^
Body fat (kg)				
T1	10.4 ± 5.57	15.8 ± 5.90	<0.001 ^1^	13.0 ± 6.33
T2	9.80 ± 5.53	14.6 ± 5.88	<0.001 ^1^	12.9 ± 6.11
*p*-value	0.217 ^2^	0.466 ^2^		0.833 ^2^
% Fat mass				
T1	12.9 ± 5.46	25.9 ± 7.26	<0.001 ^1^	19.8 ± 9.08
T2	12.9 ± 5.32	25.8 ± 7.54	<0.001 ^1^	19.4 ± 8.73
*p*-value	0.521 ^2^	0.259 ^2^		0.419 ^2^
Fat free mass (kg)				
T1	64.1 ± 7.46	42.5 ± 5.60	<0.001 ^1^	53.7 ± 12.7
T2	65.0 ± 6.96	43.1 ± 6.28	<0.001 ^1^	54.5 ± 12.8
*p*-value	0.005 ^2^	0.001 ^2^		<0.001 ^2^
Total body water (kg)				
T1	46.9 ± 5.46	31.1 ± 4.08	<0.001 ^1^	39.3 ± 9.31
T2	48.3 ± 5.67	31.5 ± 4.57	<0.001 ^1^	40.2 ± 9.86
*p*-value	<0.001 ^2^	0.001 ^2^		<0.001 ^2^

^1^ Student’s *t*-test for independent samples (*p* < 0.05); ^2^ Student’s *t*-test for paired samples (*p* < 0.05); ^3^ Pearson Chi-square (*p* < 0.05); ^4^ McNemar’s test (*p* < 0.05).

**Table 3 healthcare-12-00942-t003:** Participants’ risk of developing ED and degree of adherence to the MD at the start and end of the course.

Diagnostic Tests Used	T1	T2	*p*-Value ^1^
SCOFF—Risk of developing general ED N (%)			
Not at risk	84 (65.1)	81 (62.8)	0.398
At risk	45 (34.9)	48 (37.2)	
EAT-26—Risk of developing AN N (%)			
Not at risk	124 (96.1)	118 (91.5)	0.058
At risk	5 (3.9)	11 (8.5)	
BITE—Risk of developing BN N (%)			
Absence of compulsive behaviors	108 (83.7)	110 (85.3)	<0.001
Altered eating patterns	21 (16.3)	19 (14.7)	
MEDAS-14—Degree of adherence to the MD N (%)			
Low-moderate (low-mod)	100 (78.3)	83 (64.3)	0.008
High	29 (21.7)	46 (35.7)	
EEQ—Influence of emotions at mealtimes N (%)			
Not at all—not very emotional	69 (53.5)	76 (58.9)	0.162
(Very) Emotional	60 (46.5)	53 (41.1)	

^1^ McNemar test (*p* < 0.05).

**Table 4 healthcare-12-00942-t004:** Participants’ risk of developing general ED at T1 and T2 and associated factors.

**SCOFF Test Risk of Developing General ED (T1) (*n* = 129)**					
**Associated Factors**		**Not at Risk (*n* = 76) N (%)**	**At Risk (*n* = 53)** **N (%)**	**OR (95% CI)**	***p*-Value**
Academic Year	First year	54 (51.9)	50 (48.1)	0.147 (0.0415–0.522)	0.001 ^1^
	Fourth year	22 (88)	3 (12)		
Faculty	HS	64 (63.4)	37 (36.6)	2.31 (0.985–5.40)	0.051 ^2^
	SSH + HPS	12 (42.9)	16 (57.1)		
EQQ	Not at all—not very emotional	46 (66.7)	23 (33.3)	2.00 (0.982–4.08)	0.055 ^2^
	(Very) emotional	30 (50)	30 (50)		
**SCOFF Test Risk of Developing General ED (T2) (*n* = 129)**					
**Associated Factors**		**Not at Risk (*n* = 81) N (%)**	**At Risk (*n* = 48) N (%)**	**OR (95% CI)**	***p*-Value**
Gender	Males	49 (73.1)	18 (26.9)	2.55 (1.22–5.32)	0.012 ^2^
	Females	32 (51.6)	30 (48.4)		
Academic Year	First year	61 (58.7)	43 (41.3)	0.355 (0.124–1.02)	0.047 ^2^
	Fourth year	20 (80)	5 (20)		
MEDAS-14	Low-mod	63 (75.9)	20 (24.1)	0.921 (0.385–2.20)	<0.001 ^2^
	High	18 (39.1)	28 (60.9)		

Health and sciences (HS), social sciences and humanities (SSH), higher polytechnic school (HPS). Degree of adherence to the MD (MEDAS-14), influence of emotions at mealtimes (EEQ). OR: odds ratio. IC: 95% confidence interval. ^1^ Fisher’s exact test (*p* < 0.05). ^2^ Pearson Chi-square (*p* < 0.05).

**Table 5 healthcare-12-00942-t005:** Participants’ risk of developing AN at T1 and T2 and associated factors.

**EAT-26 Risk of Developing AN (T1) (*n* = 129)**					
**Associated Factors**		**Not at Risk (*n* = 124) N (%)**	**At Risk (*n* = 5) N (%)**	**OR (95% CI)**	***p*-Value**
Gender	Males	67 (100)	0 (0)	12.9 ^a^ (0.69–239)	0.024 ^1^
	Females	57 (91.9)	5 (8.1)		
**EAT-26 Risk of Developing AN (T2) (*n* = 129)**					
**Associated Factors**		**Not at Risk (*n* = 118) N (%)**	**At Risk (*n* = 11) N (%)**	**OR (95% CI)**	***p*-Value**
Gender	Males	65 (97)	2 (3)	5.52 (1.14–26.6)	0.026 ^1^
	Females	53 (85.5)	9 (14.5)		
MEDAS-14	Low-mod	79 (95.2)	4 (4.8)	3.54 (0.979–12.8)	0.043 ^2^
	High	39 (84.8)	7 (15.2)		

Health and sciences (HS), social sciences and humanities (SSH), higher polytechnic school (HPS). Degree of adherence to the MD (MEDAS-14), influence of emotions at mealtimes (EEQ). OR: odds ratio. IC: 95% confidence interval. ^1^ Fisher’s exact test (*p* < 0.05). ^2^ Pearson Chi-square (*p* < 0.05). ^a^ Haldane–Anscombe correction applied.

**Table 6 healthcare-12-00942-t006:** Participants’ risk of developing BN at T1 and T2 and associated factors.

**BITE Test Risk of Developing BN (T1) (*n* = 129)**					
**Associated Factors**		**Absence of Compulsive Behaviors (*n* = 109)**	**Altered Eating Patterns (*n* = 20)**	**OR (95% CI)**	***p*-Value**
		**N (%)**	**N (%)**		
Gender	Males	61 (91)	6 (9)	3.03 (1.08–8.48)	0.049 ^2^
	Females	47 (77)	14 (23)		
Faculty	HS	88 (88)	12 (12)	2.93 (1.06–8.12)	0.033 ^2^
	SSH + HPS	20 (71.4)	8 (28.6)		
EQQ	Not at all—not very emotional	64 (94.1)	4 (5.9)	5.82 (1.82–18.6)	0.001 ^1^
	(Very) Emotional	44 (73.3)	16 (26.7)		
**BITE Test Risk of Developing BN (T2) (*n* = 129)**					
**Associated Factors**		**Absence of Compulsive Behaviors (*n* = 110)**	**Altered Eating Patterns (*n* = 19)**	**OR (95% CI)**	***p*-Value**
		**N (%)**	**N (%)**		
Gender	Males	61 (91)	6 (9)	2.70 (0.955–7.61)	0.054 ^2^
	Females	49 (79)	13 (21)		
EQQ	Not at all—not very emotional	69 (90.8)	7 (9.2)	2.89 (1.05–7.91)	0.034 ^2^
	(Very) Emotional	41 (77.4)	12 (22.6)		

Health and sciences (HS), social sciences and humanities (SSH), higher polytechnic school (HPS). Degree of adherence to the MD (MEDAS-14), influence of emotions at mealtimes (EEQ). OR: odds ratio. IC: 95% confidence interval. ^1^ Fisher’s exact test (*p* < 0.05). ^2^ Pearson Chi-square (*p* < 0.05).

## Data Availability

Data described in the manuscript, code book, and analytic code will be made available upon application and approval.

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
