# Peer review of "Risk Factors for Eating Disorders in University Students: The RUNEAT Study"

_healthcare, 2024, doi:10.3390/healthcare12090942_

Round 1

Reviewer 1 Report

Comments and Suggestions for Authors

This was a prospective observational study conducted with a representative sample of 129 students of the Universidad Europea del Atlántico that evaluated the predisposition of university students to develop AN, BN, and general EDs by applying validated screening tests.   Additionally, investigators presented data regarding characteristics of students: beginning and end of year, adherence to the Mediterranean Diet and other variables including first vs 4th year, independent living, curriculum enrolled, etc.  While the study provides needed data which may help highlight characteristics of university students who are at risk of ED, however, as written, the conclusions are somewhat misleading.  As one example, the major study conclusion (abstract line 32-33) is that “the profile of the university student with the highest risk of developing EDs is a female student in her first year and studying a degree in a faculty other than health sciences.”   This is strongly worded based on an observation from a non-probability sample of volunteers which the study was not specifically designed to test (i.e., random).   It is suggested that the authors revise their Conclusions to indicate the exploratory nature of this study.  Other specific comments are below.

 Specific comments:

1.        Please clarify the purpose of this study as designed apriori.  As it is currently written, each outcome is discussed as if it were a primary outcome for which the study was specifically designed to test.  However, it actually appears to be an exploratory pilot study, conducted to generate observational data for future testing in an appropriately designed study.  Please clarify and adjust Conclusions accordingly.   

2.        For multiple hypothesis testing, it is appropriate to adjust the statistics using Bonferonni or other adjustment.    Please revise or justify why this is not possible.

3.        As written, Discussion and Conclusion are redundant and should be revised.

Author Response

Dear reviewer,

Thank you for giving us the opportunity to improve our manuscript. Here is our response. 

This was a prospective observational study conducted with a representative sample of 129 students of the Universidad Europea del Atlántico that evaluated the predisposition of university students to develop AN, BN, and general EDs by applying validated screening tests.   Additionally, investigators presented data regarding characteristics of students: beginning and end of year, adherence to the Mediterranean Diet and other variables including first vs 4th year, independent living, curriculum enrolled, etc.  While the study provides needed data which may help highlight characteristics of university students who are at risk of ED, however, as written, the conclusions are somewhat misleading.  As one example, the major study conclusion (abstract line 32-33) is that “the profile of the university student with the highest risk of developing EDs is a female student in her first year and studying a degree in a faculty other than health sciences.”   This is strongly worded based on an observation from a non-probability sample of volunteers which the study was not specifically designed to test (i.e., random).   It is suggested that the authors revise their Conclusions to indicate the exploratory nature of this study.  Other specific comments are below.

(text in blue on the manuscript) We have modified the conclusions based on the exploratory nature of the study. Furthermore, we have clarified in section 2.1 “study design” that this is an exploratory study based on an observation from a non-probability sample of volunteers to identify certain common characteristics in those students most at risk of developing ED.

 Specific comments:

  1. Please clarify the purpose of this study as designed apriori.As it is currently written, each outcome is discussed as if it were a primary outcome for which the study was specifically designed to test.  However, it actually appears to be an exploratory pilot study, conducted to generate observational data for future testing in an appropriately designed study.  Please clarify and adjust Conclusions accordingly.   

(text in blue on the manuscript) We have clarified that the study was designed to identify those characteristics common to those students most at risk of developing each of the ED studied (general ED, AN, and BN) based on the results of the SCOFF, EAT-26, and BITE screening test. Based on this, we have reworked the results section, including a section for each ED and the individual effect of each of the variables analyzed (sex, place of residence, course, faculty, and quality of diet). As you say, this initial data will be used for the design of future testing and correlation studies.

  1. For multiple hypothesis testing, it is appropriate to adjust the statistics using Bonferonni or other adjustment. Please revise or justify why this is not possible.

(text in red on the manuscript) Because most of the variables are categorical, it has been decided to include OR and CI. This allows us to establish risk or protective relationships between the variables that have been analyzed. Furthermore, the p-value is included to verify or exclude the significance of the connection between variables; p-values less than 0.05 are regarded as significant.

  1. As written, Discussion and Conclusion are redundant and should be revised.

(text in blue on the manuscript) In transcribing the article into the journal template, part of the discussion was copied into the conclusions section. Besides, we have reviewed both the discussion and the conclusion according to the new data generated. This has been corrected.

Best regards, 

Reviewer 2 Report

Comments and Suggestions for Authors

Introduction

Background to the topic has been well-described. Research gap has been highlighted. 

Research aim is not representing the main aim of the paper (Line 103-104).

Research hypothesis should be presented in paragraph format, avoiding bullets.

Some degree of improvement is required with regard to the language used. 

Materials and methods

2.1 - should be "Study design"

2.1 - the study design should be elaborated further here

Line 176 - is the bold sentence a section header?

Line 184 - .. body mass index classification. 

Line 189 - BMI was distributed using the WHO range classification is redundant 

Results

It is unclear why the authors did not consider logistic regression analysis analyzing the factors against AN, BN and general EDs, respectively. 

Lot of emphasis on sex when it comes to data presentation, and the reason behind this is not clear. 

Discussion & Conclusions

There seems to be confusion in the separation between the discussion and the conclusion. 

Overall

Check the journal's formatting guidelines. 

Comments on the Quality of English Language

There is a need to recheck the manuscript for language consistency before resubmission. 

Author Response

Dear reviewer, 

Thank you for giving us the opportunity to improve our manuscript. Here are the responses to the suggestions for improvement.

Introduction

Background to the topic has been well-described. Research gap has been highlighted. 

Research aim is not representing the main aim of the paper (Line 103-104).

(in blue in the manuscript) The objective has been modified to comply with the character of exploratory research (Line 113-115)

“The purpose of the current study is to ascertain the risk of developing general ED, AN, and BN as well as to examine the effects of sex, housing location, academic year, faculty, and dietary quality on that risk”.

Research hypothesis should be presented in paragraph format, avoiding bullets.

(in blue in the manuscript) The assumptions have been rewritten in one paragraph

Some degree of improvement is required with regard to the language used. 

(in blue in the manuscript) We have sent the article for language revision to improve the language used.

Materials and methods

2.1 - should be "Study design"

(in blue in the manuscript) We have made a typo. We have changed it to study design.

2.1 - the study design should be elaborated further here

(in blue in the manuscript) We have further developed the explanation of the study design.

Line 176 - is the bold sentence a section header?

(in blue in the manuscript)We wanted to highlight in bold the phases of the data collection:

  1. Administration of an online questionnaire
  2. Performance of anthropometric measurements

Line 184 - .. body mass index classification. 

(in blue in the manuscript)We have included the word “classification”.

Line 189 - BMI was distributed using the WHO range classification is redundant 

(in blue in the manuscript) This sentence was deleted so as not to be redundant as it was included in the previous paragraph.

Results

It is unclear why the authors did not consider logistic regression analysis analyzing the factors against AN, BN and general EDs, respectively. 

(in red in the manuscript) Because most of the variables are categorical, it has been decided to include OR and CI. This allows us to establish risk or protective relationships between the variables that have been analyzed. Furthermore, the p-value is included to verify or exclude the significance of the connection between variables; p-values less than 0.05 are regarded as significant.

Lot of emphasis on sex when it comes to data presentation, and the reason behind this is not clear. 

(in blue in the manuscript) In Table 1, we have decided to present the data divided according to sex to better categorize the different study groups. In Table 2, we have decided to present the data divided according to sex because there are differences in body composition between males and females. It should be mentioned that, among the characteristics, gender has been found to have a greater correlation with the chance of having ED. 

Discussion & Conclusions

There seems to be confusion in the separation between the discussion and the conclusion. 

(in blue in the manuscript) In transcribing the article into the journal template, part of the discussion was copied into the conclusions section. This has been corrected.

Overall

Check the journal's formatting guidelines. 

(in blue in the manuscript) We have changed and adapted the format of the headings and tables to the format requested.

There is a need to recheck the manuscript for language consistency before resubmission. 

(in blue in the manuscript) We have submitted the article for language revision.

Best regards, 

Round 2

Reviewer 1 Report

Comments and Suggestions for Authors

No additional comments. Authors addressed previous concerns.  Appears there may be formatting issue at end of Introduction? 

Author Response

Dear reviewer, 

Best regards, 

Reviewer 2 Report

Comments and Suggestions for Authors

Abstract

Line 16-18 - rephrase the sentence.

Line 19-20 - 'exploratory observational pilot study' does not reflect the paper. Why is this a pilot study? Are the authors planning a large-scale cross-sectional study? There was no qualitative component that showed this was an exploratory study either. 

Introduction

The chapter should end with a research aim. 

Methods

2.1 Should describe the study design, not the measures or analysis. 

2.4 It's unclear what test was used to generate the OR. 

Results

3.2 and so on - avoid using abbreviations on section header

Comments on the Quality of English Language

Moderate changes are required. 

Author Response

Dear reviewer, 

Best regards, 
